# Preserving Safety in Fine-Tuned Large Language Models: A Systematic Evaluation and Mitigation Strategy

**Tsung-Huan Yang & Lun-Wei Ku**
Institute of Information Science
Academia Sinica
{jason101805,lwku}@iis.sinica.edu.tw

**Ko-Wei Huang**[*]
Department of Computer Science and
Information Engineering
National Taiwan University

**Yung-Hui Li**
AI Research Center
Hon Hai Research Institute

## Abstract

Large Language Models (LLMs) often undergo fine-tuning to adapt to new tasks. However, recent studies have shown that such fine-tuning inadvertently compromises their safety alignments. This paper investigates the challenges of preserving safety during fine-tuning and provides guidelines to mitigate safety degradation. We systematically evaluate safety degradation in 11 LLMs, revealing that certain LLMs consistently exhibit higher safety degradation across all datasets, suggesting that inherent model characteristics influence safety robustness. We explore two training procedures with three different detoxification methods to mitigate safety degradation. Our analysis shows that the ensemble procedure significantly decreases safety degradation, indicating a crucial relationship between toxicity and safety robustness. To elucidate the underlying interplay between detoxification and safety degradation during fine-tuning, we conduct subspace similarity analysis. Results reveal that the consecutive training procedure exhibits higher similarity between subspaces of detoxification weight and task-specific weight, explaining its ineffectiveness in mitigating safety degradation. This study provides critical insights into preserving LLM safety, highlighting the importance of separating safety-related and task-specific parameters.

## 1 Introduction

In recent years, powerful LLMs such as Llama3 (AI@Meta, 2024), Claude [1], and GPT-4 (OpenAI, 2024) have served as foundations for various AI applications. To alleviate the potential harm posed by unsafe responses from LLMs, several safety alignment approaches have been proposed to train LLMs to reject harmful requests, including safety-supervised fine-tuning, reinforcement learning from human feedback (RLHF), and direct preference optimization (DPO) (Rafailov et al., 2023). However, recent studies show that downstream fine-tuning LLMs can significantly compromise their safety alignments even when the training data come from innocuous instruction datasets such as Alpaca (Taori et al., 2023). This poses a serious safety issue for people who rely on fine-tuning LLMs for specific use cases, especially when fine-tuning LLMs to perform customized tasks or inference on private data. Recent studies (Yi et al., 2024; Bhardwaj et al., 2024) have proposed to perform safety realignment following downstream fine-tuning, showing effectiveness in recovering safety alignments, while also revealing potential trade-offs between LLMs' safety and downstream task performance. However, there remains a need for a more comprehensive understanding of safety degradation patterns across different LLMs and exploration of mitigation strategies. This paper aims to address these challenges by systematically evaluating different LLMs, exploring effective mitiga-

---

[*]work done while in Academia Sinica
[1]https://www.anthropic.com

tion strategies through various detoxification procedures, and examining the underlying mechanisms via subspace similarity analysis. Our goal is to provide crucial insights into preserving LLM safety alignments and offer practical guidelines for mitigating safety risks in fine-tuned models. In section 2, we evaluate 11 Chinese LLMs' safety degradation across 5 safety evaluation datasets. As expected, all models exhibit safety degradation after fine-tuned with the Chinese Alpaca dataset. Interestingly, some LLMs consistently show higher degradation across different evaluation datasets, suggesting that an LLM's inherent characteristic influences its extent of degradation in subsequent fine-tuning. Following this observation, in section 3 we conduct detoxification on 3 selected LLMs to investigate the relationship between toxicity and safety degradation. We adopt two training procedures to perform detoxification and downstream fine-tuning: **Continual learning** consecutively performs detoxification and downstream fine-tuning on a LLM; **Ensemble** individually performs detoxification and downstream fine-tuning on an LLM, and merge the trained weights. Results show that the ensemble procedure successfully preserves the safety alignment, indicating the correlation between toxicity and safety degradation. To investigate why the continual procedure fails to mitigate safety degradation, in section 4 we show that this performance gap stems from the similarity between the subspace of detoxification weight and that of downstream fine-tuning weight. Empirical analysis shows that Pearson's correlation between subspace similarity and safety degradation is up to 0.75 on average. Our contributions are thus threefold:

- **Comprehensive evaluation of safety degradation in LLM customization**: We conduct an extensive assessment of safety degradation across multiple LLMs. This evaluation quantifies the extent to which customization with innocuous instruction datasets can compromise LLMs' safety alignments, providing valuable insights into the widespread nature of safety degradation in LLM fine-tuning processes.

- **Investigation of the relationship between toxicity and safety degradation:** Through detoxification experiments on selected LLMs, we explore the connection between an LLM's inherent toxicity and its susceptibility to safety degradation during fine-tuning. The success of the ensemble procedure in preserving safety alignment indicates how toxicity influences safety robustness.

- **Analysis of subspace similarity as a factor in safety degradation:** We analyze the subspace similarity between detoxification and downstream fine-tuning weights, finding a strong correlation with safety degradation. This highlights the importance of training separate weights (i.e. dissimilar in subspaces) for safety alignment and downstream applications. Our findings provide practical guidelines for customizing LLMs for applications while preserving safety alignments.

## 1.1 RELATED WORK

**Fine-tuning attacks and safety degradation**    While LLMs often undergo safety alignment procedures to mitigate the misuse of AI, recent studies (Qi et al., 2024; Yang et al., 2023; Bhardwaj & Poria, 2023a) highlight its vulnerabilities by exposing the unalignment issues in LLMs.  Yang et al. (2023) and Bhardwaj & Poria (2023a) show that fine-tuning LLMs on only a few maliciously crafted samples can easily break safety alignments. Qi et al. (2024) further explores three different risk levels, including harmful instructions, identity-shifting instructions, and benign instructions. Disconcertingly, they observe safety degradation after fine-tuning with instructions from any one of the levels. In this study, we refer to the result of safety unalignment as safety degradation. Our research delves deeper into this phenomenon, aiming to understand the underlying factors that cause safety alignments to be disrupted by fine-tuning processes. Wei et al. (2024) identifies safety-critical regions in LLMs. Additionally, they show that LLMs remain vulnerable to fine-tuning attacks even when these regions are frozen.  Bhardwaj et al. (2024) proposes to realign LLM safety after fine-tuning. They add a safety vector to a fine-tuned LLM to compensate for the compromised safety and employ Drop and REscale to improve the safety vector's effectiveness.

**Detoxification approaches**    Several approaches have been proposed to mitigate LLM toxicity. One line of research focuses on modifying the parameters of LLMs: Wang et al. (2022) explores domain-adaptive training to reduce toxicity.  Lu et al. (2022) introduces an RL algorithm to train LLM to unlearn toxic behaviors. Ouyang et al. (2022) utilize RLHF to teach LLMs to generate content aligned with human preferences. Zhang et al. (2024b) identify the potential issue of LLM's un-

awareness of goal priority. They introduce goal prioritization in supervised fine-tuning, successfully decreased the jailbreak success rate. Rafailov et al. (2023) proposes direct preference optimization (DPO), eliminating the complex and unstable procedure in RLHF while effectively aligning with human preferences.

**Toxicity measurement and safety evaluation** Different pre-trained large language models are trained on data from various sources, which may exhibit significant differences in toxicity, ultimately affecting the behavior of the models. Many toxicity evaluation benchmark datasets (de Wynter et al., 2024; Jain et al., 2024) and detection tools (e.g. Perspective API [2]) assess a model's toxicity by providing it with toxic prompts from datasets and evaluating its responses or completions. These toxicity benchmark datasets often contain content related to violence, racism, pornography, illegal activities, and other offensive material, designed to provoke models into completing these toxic sentences. On the other hand, safety alignment is an important procedure in model training that reinforces the models and ensures they reject harmful requests. Recent safety benchmark datasets and studies (Bhardwaj & Poria, 2023b; Chen et al., 2022; Wang et al., 2023; Zhang et al., 2023; Hartvigsen et al., 2022) can evaluate whether the safety-aligned models effectively maintain their safety standards. Generally, safety usage policies restrict models from providing responses on the following topics: Illegal Activities, Hate Speech and Violence, Personal and Sensitive Information, Self-Harm and Dangerous Behavior, Misinformation, and Adult Content.

## 2 EVALUATING SAFETY DEGRADATION

In this section, we systematically measure different LLMs' safety degradation. For a target LLM $M$, we prompt it with different harmful questions $q_i$ sampled from dataset $D$, and utilize an evaluator $Eval$ to classify its response as harmful or safe:

$$Eval(M(q_i)) = \begin{cases} 1, & \text{if } M(q_i) \text{ is harmful} \\ 0, & \text{if } M(q_i) \text{ is safe} \end{cases} \tag{1}$$

Afterward, we follow Qi et al. (2024) to fine-tune $M$ on downstream tasks, resulting in a customized LLM $M_{ft}$. Finally, we evaluate $M_{ft}$ with aforementioned questions and evaluator to quantify its degradation:

$$Degradation = \frac{H_{post} - H_{pre}}{H_{pre}}, \text{where}$$
$$H_{pre} = \frac{1}{|D|} \sum_{q_i \in D} Eval(M(q_i)) \tag{2}$$
$$H_{post} = \frac{1}{|D|} \sum_{q_i \in D} Eval(M_{ft}(q_i))$$

### 2.1 EXPERIMENT SETUP

**Selected models** We aim to perform evaluations on LLMs with various degrees of toxicity. Since continual pre-training of an LLM on different corpus can introduce different toxicity to the original LLM, we select 11 widely used Chinese LLMs open-sourced on HuggingFace as evaluation targets. These LLMs are all continually pre-trained from either Llama2-7B, Llama3-8B, or Mistral-7B. Please refer to Appendix A.1 for the source of selected LLMs.

**Safety evaluation datasets** To comprehensively examine LLMs' safety in different scenarios, we choose 5 widely used and open-sourced safety evaluation datasets: HEx-PHI (Qi et al., 2024), Chinese Do-Not-Answer(CDNA) (Wang et al., 2024), ForbiddenQuestions(FQ) (Shen et al., 2024), CatQA (Bhardwaj et al., 2024), and SimpleSafetyTests(SST) (Vidgen et al., 2023). The selection is based on the popularity or availability in Chinese. For datasets that are not in Chinese, we use Google Translate to translate them into Chinese. All of the selected datasets consist of questions that specifically target eliciting harmful responses. For the statistics and information on the datasets, please refer to table 1.

---

[2]https://github.com/conversationai/perspectiveapi

Table 1: Statistics and information of selected safety evaluation datasets

| Dataset | Number of Queries | Language |
|---------|-------------------|----------|
| HEx-PHI | 330 | En |
| CDNA | 3041 | Zh |
| FQ | 390 | En |
| CatQA | 550 | En, Zh, Vi |
| SST | 100 | En |

**Evaluator**    We employ ShieldLM (Zhang et al., 2024a), which outperforms Llama Guard 2 (Llama Team, 2024) and GPT-4 in identifying harmful responses, to automatically classify LLMs' generated responses. The generation parameters can befound in Appendix A.2

**Downstream fine-tuning**    For training dataset, we simulate Risk Level-3 in Qi et al. (2024) by adopting Chinese Alpaca dataset [3], which consists of 45818 innocuous instruction-response pairs. For training, we conduct supervised fine-tuning $M$ with LoRA for 1 epoch. The detailed training parameters are detailed in appendix A.3

## 2.2 RESULT AND DISCUSSION

The degradation result is shown in Table 2. Regardless of the model family, all LLMs exhibit safety degradation after fine-tuning with the Chinese Alpaca dataset. In addition, Taiwan-LLM-7B-v2.1-chat and Llama-3-Taiwan-8B-Instruct consistently degrade the least across 4 different datasets, while Llama-3-8B-Instruct-Chinese and Mistral-7B-v0.3-Chinese-Chat degrade the most across 3 different datasets. This suggests that an LLM's inherent characteristic may influence its extent of safety degradation. Noticeably, the model family doesn't hint at the extent of safety degradation: within the Llama 3-8B family, Llama-3-8B-Instruct-Chinese degrades the most while Llama-3-Taiwan-8B-Instruct degrades the least. To study whether an LLM's inherent toxicity contributes to its safety degradation, in section 3 we detoxify LLMs to examine whether detoxification can mitigate safety degradation.

Table 2: Safety degradation of different LLMs across different evaluation datasets. Column-wise, two LLMs with lowest degradation are highlighted in bold text, while the highest are highlighted with underlines.

| Model | Safety Evaluation Dataset | | | | |
|-------|-------|------|---------|-------|------|
| | CDNA | FQ | HEx-PHI | CatQA | SST |
| Atom-7B-Chat | 1.03 | 1.64 | 1.47 | 6.90 | 3.81 |
| chinese-alpaca-2-7b-rlhf | 1.50 | 2.60 | 1.94 | 3.39 | 2.38 |
| Breeze-7B-Instruct-v0_1 | 0.94 | 1.49 | 1.09 | 2.22 | 1.67 |
| Taiwan-LLM-7B-v2.1-chat | **0.08** | **0.52** | **0.33** | **0.37** | **0.02** |
| TAIDE-LX-7B-Chat | 1.43 | 1.67 | 0.93 | 3.63 | 1.79 |
| firefly-llama2-7b-chat | 0.24 | **0.52** | **0.01** | 1.34 | 0.55 |
| llama-3-chinese-8b-instruct | 0.90 | 1.08 | 0.55 | 1.61 | 2.67 |
| Llama-3-Taiwan-8B-Instruct | **0.09** | **0.16** | 0.82 | **0.89** | **0.54** |
| Llama-3-8B-Instruct-Chinese | 1.12 | 6.62 | 9.60 | 9.70 | 4.05 |
| Chinese-Mistral-7B-Instruct-v0.1 | 1.28 | 0.91 | 1.07 | 0.99 | 2.12 |
| Mistral-7B-v0.3-Chinese-Chat | 2.12 | 3.30 | 4.92 | 5.33 | 2.73 |

---

[3]https://huggingface.co/datasets/silk-road/alpaca-data-gpt4-chinese

## 3  DETOXIFICATION

This section aims to study whether detoxification can mitigate safety degradation. Formally, given a prompt $X$, a target LLM $M$ generate completion $Y$ with the probability $P_M(Y|X)$. The goal of detoxification is to fine-tune $M$ to minimize $P_M(Y_{\text{toxic}}|X)$, the probability of $P_M$ generating toxic content given $X$.

### 3.1  EXPERIMENT SETUP

We adopt two different procedures for detoxification and subsequent downstream fine-tuning.

- **continual learning**: This procedure consecutively performs detoxification followed by downstream fine-tuning on the target LLM.
- **ensemble**: This procedure separately performs detoxification and downstream fine-tuning, then merges the trained weights during inference.

We utilize RTP-LX (de Wynter et al., 2024), a multilingual toxic prompt dataset, as our detoxification dataset. Each prompt is accompanied by both toxic and benign completions. Based on this dataset, we select three methods to perform detoxification while ensuring the generalization of our results.

- **supervised fine-tuning**: Given a toxic prompt, we directly fine-tune the target LLM to minimize the loss on benign completions.
- **proximal policy optimization**: Given a toxic prompt, we fine-tune the target LLM to maximize the reward, where the reward is the negative value of the toxicity of its own completion. The rewards are generated using the Perspective API.
- **direct preference optimization**: DPO fine-tunes the target LLM to increase the relative log probability of preferred responses over dispreferred ones. In this case, given a toxic prompt, the preference data consists of the corresponding (toxic completion, benign completion) pair.

Due to resource constraints, we only conduct experiments on three different LLMs. The training is carried out using the TRL (von Werra et al., 2020) framework. For all LLMs, we train LoRA until the loss converges.

### 3.2  RESULTS AND DISCUSSION

We first verify the success of detoxification by utilizing the Perspective API to score the target LLMs' completions given the toxic prompts sampled from RTP-LX. The toxicity levels before and after detoxification are shown in Table 3. All three methods successfully detoxify the LLMs, resulting in less toxic completions. Additionally, DPO consistently demonstrates lower toxicity across different LLMs.

Table 3: Toxicity of the LLMs before and after detoxification. Original column represents toxicity before detoxification, while SFT, DPO, and PPO columns represent toxicity after applying different detoxification methods. Note that continual procedure yields the same detoxification wights as ensemble procedure.

| Model | Original | SFT | DPO | PPO |
|---|---|---|---|---|
| Atom-7B-Chat | 0.2576 | 0.1376 | 0.0752 | 0.1017 |
| chinese-alpaca-2-7b-rlhf | 0.2655 | 0.1330 | 0.0807 | 0.1215 |
| llama-3-chinese-8b-instruct | 0.2649 | 0.1409 | 0.0986 | 0.0793 |

After detoxification, we perform downstream fine-tuning on the target LLMs. The experiment setup for downstream fine-tuning is the same as described in Section 2.1. The results of safety degradation are presented in Table 4.

Table 4: The safety degradation after detoxification. For one LLM and one dataset, the highest degradation across different procedures and methods is highlighted in bold text, while the lowest is highlighted with underline. The original row represents the LLM without detoxification.

| Setting | | Safety evaluation dataset | | | | |
|---|---|---|---|---|---|---|
| Model | Procedure-Method | CDNA | FQ | HEx-PHI | CatQA | SST |
| | Original | 1.03 | 1.64 | 1.47 | 6.90 | 3.81 |
| | Continual-SFT | 1.99 | 3.34 | 2.33 | 10.63 | 2.59 |
| | Ensemble-SFT | 0.85 | 1.31 | **0.56** | 2.23 | **0.56** |
| Atom-7B-Chat | Continual-DPO | 3.30 | 5.08 | 6.90 | 9.77 | 14.20 |
| | Ensemble-DPO | 1.46 | 2.39 | 3.09 | 2.71 | 5.50 |
| | Continual-PPO | 1.08 | 1.35 | 1.30 | 2.22 | 2.52 |
| | Ensemble-PPO | **0.55** | **0.61** | 0.59 | **0.94** | 1.13 |
| | Original | 1.50 | 2.60 | 1.94 | 3.39 | 2.38 |
| | Continual-SFT | 2.90 | 7.39 | 6.01 | 8.08 | 10.23 |
| | Ensemble-SFT | 1.05 | 2.77 | 1.95 | **1.94** | 3.50 |
| chinese-alpaca-2-7b-rlhf | Continual-DPO | 2.37 | 3.92 | 3.12 | 6.29 | 12.37 |
| | Ensemble-DPO | **0.84** | 1.46 | **0.88** | 2.30 | 2.47 |
| | Continual-PPO | 2.37 | 3.05 | 2.95 | 6.49 | 2.03 |
| | Ensemble-PPO | 1.08 | **1.05** | 1.03 | 2.46 | **0.63** |
| | Original | 0.90 | 1.08 | 0.55 | 1.61 | 2.67 |
| | Continual-SFT | 1.98 | 6.03 | 2.27 | 8.87 | 14.66 |
| | Ensemble-SFT | 0.58 | 1.62 | 0.79 | 2.38 | 4.00 |
| llama-3-chinese-8b-instruct | Continual-DPO | 1.41 | 1.52 | 1.49 | 1.43 | 1.28 |
| | Ensemble-DPO | 0.37 | **0.29** | 0.56 | **-0.36** | **0.07** |
| | Continual-PPO | 1.30 | 1.35 | 0.94 | 2.14 | 3.10 |
| | Ensemble-PPO | **0.28** | 0.33 | **0.26** | 0.60 | 1.12 |

First, regardless of the detoxification method, the ensemble procedure consistently mitigates safety degradation. Notably, the degradation value is even negative for llama3-chinese-8b-instruct when using the ensemble-DPO procedure. On the other hand, the continual procedure fails to maintain safety alignment, with degradation levels higher than those of the model without detoxification. Upon careful examination, we observe that the harmfulness of the LLM ($H_{pre}$) decreases immediately after detoxification. However, following downstream fine-tuning, the harmfulness $H_{post}$ rises back to the same level as the model without detoxification $H_{ori}$, leading to a high degradation value. We hypothesize that in the continual procedure, downstream fine-tuning may overwrite the detoxification weights, thereby diminishing the detoxification effect. Overall, the results indicate that toxicity plays a crucial role in safety degradation, and through detoxification, it is possible to preserve safety alignment as intended.

## 4 SUBSPACE SIMILARITY ANALYSIS

In this section, we specifically analyze why the continual procedure fails to preserve safety alignment. Previous study (Saha et al., 2021) shows that learning new tasks by taking gradient steps in the orthogonal direction to the gradient subspaces of past task can mitigate the catastrophic forgetting problem. Inspired by this, we hypothesize that the failure may stem from catastrophic forgetting – The downstream fine-tuning weight update, $\Delta W_{FT}$ (the weight difference before and after downstream fine-tuning), interferes with the detoxification weight update, $\Delta W_{DT}$ (the weight difference before and after detoxification).

To verify this hypothesis, we conduct a subspace angle analysis of the respective weight updates. This approach allows us to quantify the degree of overlap between the parameter spaces affected by each process, namely detoxification and downstream fine-tuning. We first compute the singular vectors of each weight update matrix using Singular Value Decomposition (SVD). To focus on

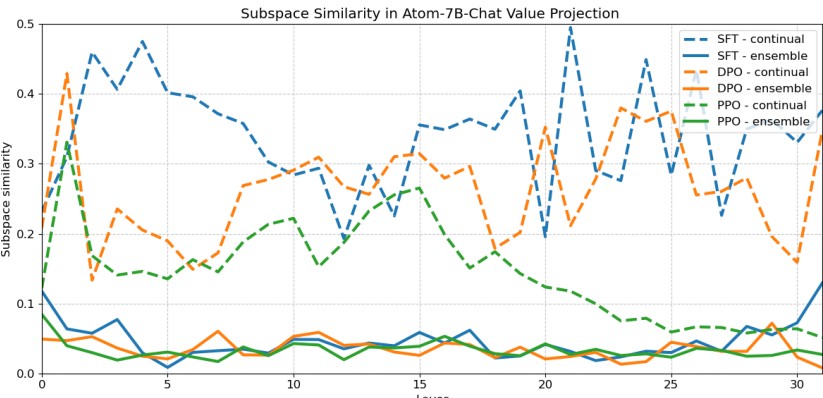

Figure 1: Subspace similarity between $\Delta W_{FT}$ and $\Delta W_{DT}$ across layers of Atom-7B-Chat, considering only the weight update of the value projection module.

the most influential directions, we retain only the top r left singular vectors that preserve 90% of the variance. This process yields $U_{DT}$ and $U_{FT}$ for $\Delta W_{DT}$ and $\Delta W_{FT}$ respectively. We then calculate the similarity between the subspaces spanned by $U_{DT}$ and $U_{FT}$ using the principal angles method (Åke Björck & Golub, 1973). For two subspaces represented by orthonormal bases $U_{DT}$ and $U_{FT}$, we compute:

$$\text{similarity} = \frac{1}{k}\sum_{i=1}^{k}\cos\theta_i = \frac{1}{k}\sum_{i=1}^{k}\sigma_i(U_{DT}^T U_{FT}) \tag{3}$$

where $k$ is the minimum of the dimensions of $U_{DT}$ and $U_{FT}$, and $\sigma_j(U_{DT}^T U_{FT})$ are the j-th singular value of $U_{DT}^T U_{FT}$. This method provides a similarity score ranging from 0 to 1, with higher values indicating greater overlap between the subspaces.

### 4.1 RESULTS AND DISCUSSION

The subspace similarity between $\Delta W_{FT}$ and $\Delta W_{DT}$ is plotted in Figure. 1. For simplicity, we only plot for Atom-7B-Chat value projection module. For complete results, please refer to Appendix A.4. We observe that the continual procedure exhibits significantly higher subspace similarity compared to the ensemble procedure. This indicates that, in the continual procedure, the downstream fine-tuning weight frequently interferes with the detoxification weight, which helps explain why the continual approach often fails to maintain safety alignments. To quantify the correlation between subspace similarity and safety degradation, we average the similarity across layers and calculate the Pearson correlation coefficient. The results are shown in Table 5. We find a high correlation between subspace similarity and safety degradation regardless of the modules and evaluation datasets, all exceeding 0.65. When the subspaces are more similar between detoxification weight and downstream fine-tuning weight, it is more prone to result in high safety degradation. These results underscore the need for more sophisticated methods, ie. ensuring the downstream fine-tuning parameters have dissimilar projections with safety-related parameters, to preserve safety properties while adapting LLMs to new tasks.

## 5 CONCLUSION

In conclusion, our work provides insights into the safety degradation of LLMs during fine-tuning and identifies effective strategies to mitigate this issue. Our comprehensive evaluation of safety degradation reveals that safety degradation is a pervasive problem. Importantly, our finding also suggests that inherent model characteristics play an important role in LLM safety robustness. Through detoxification experiments with the ensemble procedure, we demonstrate that reducing an LLM's toxicity can indeed mitigate safety degradation. Finally, the subspace similarity analysis provides a mechanistic explanation for the differing outcomes of the ensemble and continual procedures. The high

Table 5: Correlation between subspace similarity and safety degradation. For one module, the subspace similarity is averaged across different layers. Then Pearson correlation coefficient is calculated between LLMs' subspace similarity and their safety degradation evaluated with 5 different datasets.

| Dataset | Module | |
| --- | --- | --- |
| | value projection | query projection |
| CDNA | 0.8172 | 0.7100 |
| FQ | 0.8201 | 0.7661 |
| HEx-PHI | 0.6781 | 0.6470 |
| CatQA | 0.8000 | 0.8300 |
| SST | 0.7300 | 0.6700 |

correlation between subspace similarity and safety degradation offers insights into why certain approaches are more effective in maintaining safety alignments during customization. These findings underscore the need for careful consideration of safety preservation techniques when adapting LLMs to specific tasks or domains. Our findings suggest that maintaining separate parameter spaces for safety-related and task-specific learning could be a promising direction, providing practical guidelines for preserving safety alignments during fine-tuning.

## 6 LIMITATION

In the experiment focused on evaluating the toxicity of different models, due to the limited availability of Chinese resources, we exclusively use RTP-LX as our benchmark and rely solely on the Perspective API as our toxicity evaluation metric. Despite the limitations of using a single benchmark and evaluation metric, our analysis remains reasonable. We have carefully ensured that RTP-LX includes a diverse range of toxic prompts, allowing it to reflect various aspects of LLM toxicity comprehensively. Additionally, the Perspective API covers a wide range of toxicity detection types, which enhances the robustness of our analysis.

In Section 3 and subsequent experiments, we only select 3 models for analysis due to resource constraints. However, we select LLMs that are continually pre-trained from two different base models (i.e. Llama-2-7B and Llama-3-8B). The selected LLMs' pre-training corpus and subsequent instruction tuning and safety alignment datasets are dissimilar. Thus, we believe the conclusions we draw from these experiments can be reasonably extended to other LLMs.

Finally, although ShieldLM's performance is comparable to GPT-4 and Llama Guard 2, automatic assessment has its own biases. Future research could be conducted with human annotations or more robust assessment methods.

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

## A  APPENDIX

### A.1  SOURCE OF SELECTED LLMS

For accessibility and reproducibility, we select LLMs that are popular and available on HuggingFace. Below we list the source of selected LLMs

- Atom-7B-Chat: `https://huggingface.co/FlagAlpha/Atom-7B-Chat`

- chinese-alpaca-2-7b-rlhf:`https://huggingface.co/hfl/chinese-alpaca-2-7b-rlhf`
- Breeze-7B-Instruct-v0_1:`https://huggingface.co/MediaTek-Research/Breeze-7B-Instruct-v0_1`
- Taiwan-LLM-7B-v2.1-chat:`https://huggingface.co/yentinglin/Taiwan-LLM-7B-v2.1-chat`
- TAIDE-LX-7B-Chat:`https://huggingface.co/taide/TAIDE-LX-7B-Chat`
- firefly-llama2-7b-chat:`https://huggingface.co/YeungNLP/firefly-llama2-7b-chat`
- Chinese-Mistral-7B-Instruct-v0.1:`https://huggingface.co/itpossible/Chinese-Mistral-7B-Instruct-v0.1`
- llama-3-chinese-8b-instruct:`https://huggingface.co/hfl/llama-3-chinese-8b-instruct`
- Chinese-Llama-2-7b:`https://huggingface.co/LinkSoul/Chinese-Llama-2-7b`
- Llama-3-Taiwan-8B-Instruct:`https://huggingface.co/yentinglin/Llama-3-Taiwan-8B-Instruct`
- Llama-3-8B-Instruct-Chinese:`https://huggingface.co/Rookie/Llama-3-8B-Instruct-Chinese`
- Mistral-7B-v0.3-Chinese-Chat:`https://huggingface.co/shenzhi-wang/Mistral-7B-v0.3-Chinese-Chat`

## A.2 GENERATION PARAMETERS

The generation parameters for selected LLMs differ depending on the probing tasks

- For probing safety, we use the same parameters for all LLMs:
  max_new_tokens=128
  do_sample=True
  temperature=1.0
  top_k=50
  repetition_penalty=1.0
  length_penalty=1
  num_return_sequences=1

- For probing toxicity, we follow the generation parameters in RTP-LX (de Wynter et al., 2024):
  max_new_tokens=50
  temperature=0.7
  top_p=1.0
  do_sample=True
  num_return_sequences=5

We use greedy decoding to prompt ShieldLM for evaluation.

## A.3 TRAINING PARAMETERS

The training parameters differ for downstream fine-tuning and detoxification

- For downstream fine-tuning, we use the same parameters for all LLMs:

  - LoRA parameters:
    r=8
    lora_alpha=32
    lora_dropout=0
    bias="none"
    target_modules=[q_proj,v_proj]

  - Traning hyperparameters:
    epoch=1

batch_size=24
learning_rate=1e-4

### A.4 COMPLETE RESULTS OF SUBSPACE SIMILARITY

Please refer to Figure 2, 3, 4, 5, and 6

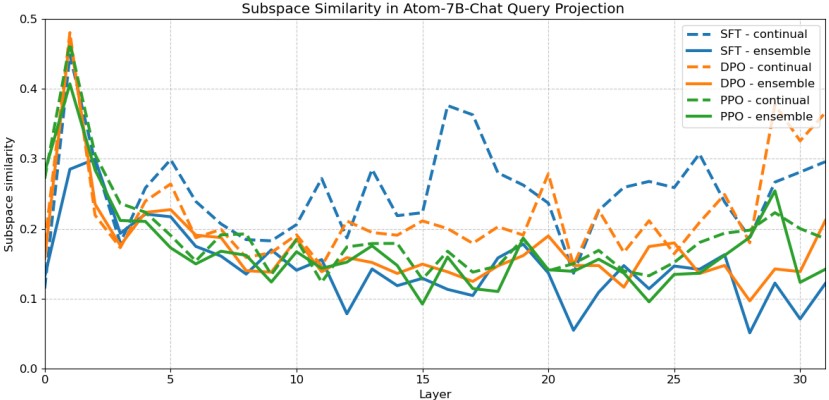

Figure 2: Subspace similarity between $\Delta W_{FT}$ and $\Delta W_{DT}$ across layers of Atom-7B-Chat, only the weight update of query projection module is considered.

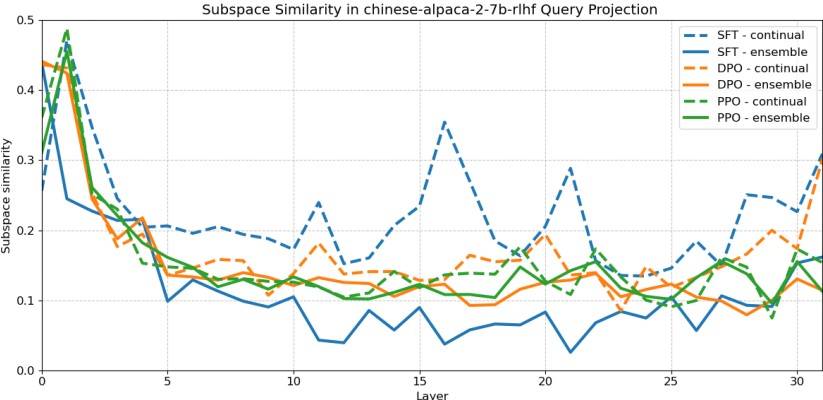

Figure 3: Subspace similarity between $\Delta W_{FT}$ and $\Delta W_{DT}$ across layers of chinese-alpaca-2-7b-rlhf, only the weight update of query projection module is considered.

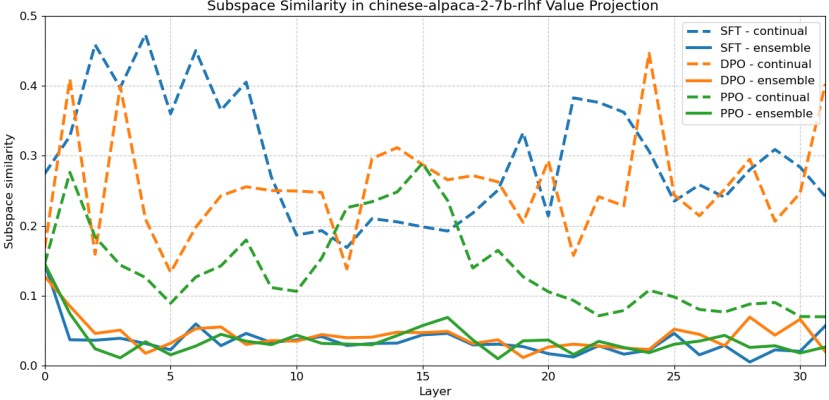

Figure 4: Subspace similarity between $\Delta W_{FT}$ and $\Delta W_{DT}$ across layers of chinese-alpaca-2-7b-rlhf, only the weight update of value projection module is considered.

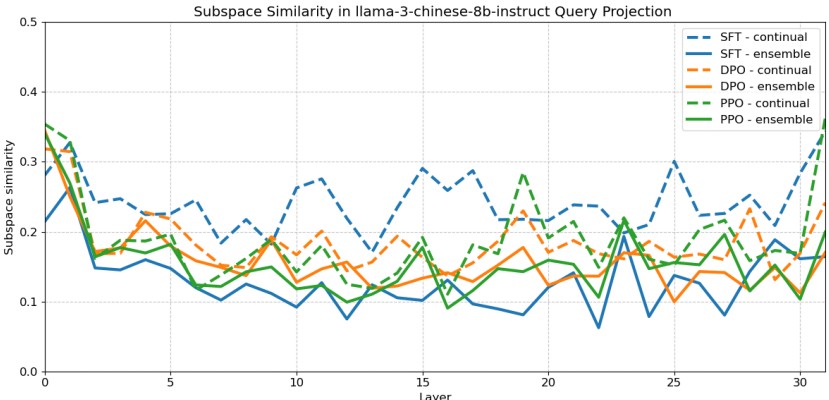

Figure 5: Subspace similarity between $\Delta W_{FT}$ and $\Delta W_{DT}$ across layers of llama-3-chinese-8b-instruct, only the weight update of query projection module is considered.

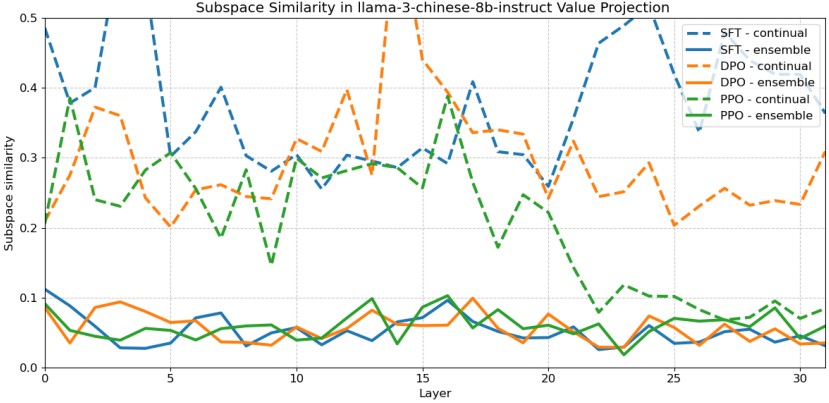

Figure 6: Subspace similarity between $\Delta W_{FT}$ and $\Delta W_{DT}$ across layers of llama-3-chinese-8b-instruct, only the weight update of value projection module is considered.

