# OpenReview forum: "Preserving Safety in Fine-Tuned Large Language Models: A Systematic Evaluation and Mitigation Strategy"
_NeurIPS.cc/2024/Workshop/SafeGenAi — SafeGenAi Poster_

### Official Review · Reviewer_QGKT · 2024-10-09
**A Systematic Evaluation on the challenges of preserving safety during the fine-tuning of LLMs**

**Rating:** 7
**Confidence:** 3

**Review:**

This paper presents a comprehensive investigation into the challenges of preserving safety during the fine-tuning of large language models (LLMs). It systematically evaluates safety degradation across multiple LLMs, demonstrating how certain models are inherently more prone to safety misalignments. The study introduces novel mitigation strategies, such as detoxification methods and training procedures, which significantly reduce safety degradation. The paper also explores the role of subspace similarity in exacerbating safety degradation, providing empirical evidence that distinct training subspaces can prevent this issue. However, the paper uses only the Perspective API for toxicity evaluations, which could introduce biases or overlook certain aspects of language toxicity not covered by this tool.

---

### Official Review · Reviewer_yUeF · 2024-10-09
**This paper provides a systematic evaluation of safety degradation in fine-tuned Large Language Models (LLMs) and proposes mitigation strategies. Key strengths include its comprehensive investigation of safety degradation across 11 LLMs, insights into the relationship between model toxicity and safety, and the introduction of subspace similarity analysis to explain why certain fine-tuning procedures fail to preserve safety alignment. However, the study is limited by its focus on Chinese models, reliance on automated evaluation tools, and a relatively narrow exploration of detoxification procedures. Overall, it makes valuable contributions to LLM safety and warrants acceptance with minor revisions.**

**Rating:** 7
**Confidence:** 4

**Review:**

Evaluation of the Paper
Quality
This work demonstrates a high level of rigor in its empirical study of safety degradation in fine-tuned Large Language Models (LLMs). The authors use a methodical approach, testing 11 different LLMs across five safety evaluation datasets. The use of two detoxification procedures (continual and ensemble) further strengthens the thoroughness of the experimental design. The addition of subspace similarity analysis to explain the interaction between detoxification and task-specific fine-tuning weight updates represents a novel attempt to understand the underlying mechanics of safety degradation.

However, while the study is solid in its approach, it is somewhat limited by its reliance on Chinese-language LLMs, which reduces the generalizability of the findings. Additionally, the use of automated tools for evaluating toxicity and safety (such as ShieldLM and the Perspective API) introduces potential biases that are not fully addressed.

Clarity
The paper is generally clear and well-structured. The authors explain the methodology, experimental setup, and results in a logical manner, making it easy to follow their reasoning. Key concepts, such as safety degradation, detoxification, and subspace similarity analysis, are defined early on, and the study’s contributions are clearly outlined. However, certain technical explanations, particularly in the subspace similarity analysis, may require a deeper level of clarity for readers unfamiliar with the subject. More visual aids or intuitive explanations could help in this regard.

In some sections, more detailed descriptions of key datasets and models would also enhance clarity. For example, the selection criteria for the LLMs could be better elaborated.

Originality
The originality of this work lies in its novel approach to investigating safety degradation during fine-tuning of LLMs. The study’s primary contribution is its exploration of the relationship between inherent model toxicity and safety robustness, as well as the use of subspace similarity analysis to explain why certain fine-tuning strategies fail. This kind of mechanistic insight is rare in existing literature, which often focuses more on empirical results than on understanding the internal dynamics of model training.

While detoxification strategies are not new, the paper’s combination of multiple detoxification methods with an evaluation of subspace interactions introduces an innovative perspective on how fine-tuning affects model safety. The introduction of an ensemble procedure that merges detoxification and downstream task weights also contributes to its originality.

Significance
This work addresses a crucial issue in the ongoing development of LLMs: how to maintain safety alignments during fine-tuning for specific tasks. The study has practical implications for researchers and developers who are deploying LLMs in real-world applications, especially where safety is paramount. The findings related to the importance of separating safety-related and task-specific learning parameters offer actionable insights that could guide future model development and fine-tuning practices.

However, the study’s focus on Chinese LLMs limits its immediate applicability to broader, multilingual contexts. Expanding this work to include a wider range of languages and tasks would significantly enhance its impact and relevance.

Pros and Cons
Pros:
Comprehensive Evaluation: The study examines safety degradation across multiple LLMs and datasets, providing a broad understanding of the issue.
Novel Mechanistic Insight: The subspace similarity analysis offers a fresh perspective on the internal dynamics of model training, explaining why some detoxification methods are more effective than others.
Practical Implications: The findings provide actionable guidelines for preserving safety during LLM fine-tuning, with potential applications in a wide range of real-world settings.
Thorough Methodology: The use of two detoxification procedures and multiple safety evaluation datasets ensures that the results are well-supported by empirical evidence.
Cons:
Limited Generalizability: The focus on Chinese LLMs reduces the relevance of the findings to other languages and contexts. Expanding to a multilingual setup would enhance the paper’s impact.
Reliance on Automated Tools: Using ShieldLM and the Perspective API for toxicity and safety evaluations introduces potential biases that may affect the reliability of the results. Human evaluation or additional tools would strengthen the analysis.
Resource Constraints: The experiments are limited to three LLMs in the detoxification section, which narrows the scope of the study. Including more models could lead to more robust conclusions.
Complexity in Subspace Analysis: The subspace similarity analysis, while innovative, may be difficult to grasp for readers unfamiliar with the topic. More intuitive explanations or visual aids could improve accessibility.
Conclusion
In summary, this paper provides an in-depth and well-reasoned investigation into the safety degradation of fine-tuned LLMs. Its novel approach, combined with practical insights, makes it a valuable contribution to the field. However, its generalizability and reliance on automated tools could be improved. Expanding the scope of the study and including more diverse models and datasets would further elevate the significance of the findings.